# Effect of Essential Oils of *Apiaceae*, *Lamiaceae*, *Lauraceae*, *Myrtaceae*, and *Rutaceae* Family Plants on Growth, Biofilm Formation, and Quorum Sensing in *Chromobacterium violaceum*, *Pseudomonas aeruginosa*, and *Enterococcus faecalis*

**DOI:** 10.3390/microorganisms11051150

**Published:** 2023-04-28

**Authors:** Patrizia D’Aquila, Giada Sena, Michele Crudo, Giuseppe Passarino, Dina Bellizzi

**Affiliations:** Department of Biology, Ecology and Earth Sciences, University of Calabria, 87036 Rende, Italy

**Keywords:** MIC, MBC, essential oils, nutrition, antimicrobials, biofilm, quorum sensing, epigenetics, adenine methylation, cytosine methylation

## Abstract

The biological role played by essential oils extracted from aromatic plants is progressively being recognized. This study evaluated the potential antibacterial activity of ten essential oils against *Chromobacterium violaceum*, *Pseudomonas aeruginosa*, and *Enterococcus faecalis* by measuring their minimum inhibitory concentration. We found that essential oils exert different antimicrobial effects, with *Origanum vulgare* and *Foeniculum vulgare* demonstrating the most significant inhibitory effect on bacterial growth for *C. violaceum* and *E. faecalis*. The growth of *P. aeruginosa* was not affected by any essential oil concentration we used. Sub-inhibitory concentrations of essential oils reduced in *C. violaceum* and *E. faecalis* biofilm formation, violacein amount, and gelatinase activity, all of which are biomarkers of the Quorum Sensing process. These concentrations significantly affect the global methylation profiles of cytosines and adenines, thus leading to the hypothesis that the oils also exert their effects through epigenetic changes. Considering the results obtained, it is possible that essential oils can find a broad spectrum of applications in counteracting microbial contamination and preserving sterility of surfaces and foods, as well as inhibiting microbial growth of pathogens, alone or in combination with traditional antibiotics.

## 1. Introduction

The evolution and dissemination of resistance factors within bacterial populations is increasingly affirming itself. Due to the massive and uncontrolled clinical overprescription of antibiotics and their misuse by the food industry, there is an increasing need to research new compounds capable of inhibiting microbial growth. In this context, products of natural origin are showing encouraging results, with a plethora of studies demonstrating that plant extracts and essential oils (EOs) provide a huge range of complex compounds possessing antifungal, antibacterial, and antiviral properties, to the point that natural additives are spreading in the industrial field for the shelf-life extension of food products [1,2,3,4,5]. Therefore, great attention is being paid to the chemical characterization of EOs and to screening their biological activity [6,7,8]. Next to the bactericidal effects exerted by many EOs, the capacity of some EOs to interfere in Quorum Sensing (QS) processes and biofilm formation has been recently documented [9,10,11]. Several studies have reported that *Origanum vulgare* oil, with its main bioactive constituents, exhibited strong activity against Gram-positive and negative species such as *Staphylococcus aureus*, *Bacillus subtilis*, *E. faecalis*, *Escherichia coli*, and other pathogenic microorganisms [12,13]. More recently, Merghni et al. demonstrated its anti-biofilm, anti-swarming, and anti-QS capabilities against pathogenic bacteria such as *S. aureus* and *Pseudomonas aeruginosa* [14]. Similarly, a reduction in the biofilm formation was observed in several bacterial strains, such as *E. coli*, *S. aureus*, *B. subtilis*, and *Candida albicans*, treated with the EO extracted from peppermint [15,16,17]. Additionally, the effects of many other EOs, including those obtained from clary sage, juniper, lemon, and marjoram, and their primary components, was found to significantly inhibit the formation of bacterial and yeast biofilms and QS [18].

Although the antibacterial properties of EOs and their main components have been examined in several studies, many issues remain to be clarified about the mechanism of action by which they perform their activities. Therefore, in this study, we evaluated the potential antimicrobial activity and the inhibitory effect on biofilm formation and the QS process of ten EOs extracted from aromatic plants grown in Calabria (Southern Italy) on the Gram-negative *Chromobacterium violaceum* and *P. aeruginosa* and the Gram-positive *E. faecalis*. These aromatic plants are grouped into *Apiaceae*, *Lamiaceae*, *Lauraceae*, *Myrtaceae*, and *Rutaceae* families. *Apiaceae* is one of the most numerous plant families. It comprises several economically important vegetables, herbs, and spices. Umbelliferous crops such as anise, fennel, carrot, coriander, and parsley are appreciated sources of botanical flavoring agents and fragrances. *Apiaceae* plants produce a wide variety of specialized metabolites, including volatile phenylpropanoids, furanocoumarins, sesquiterpene coumarins, polyacetylenes, and phthalides, some of which are considered uncommon natural phytochemicals distinctive of the family [19]. *Lamiaceae,* or the mint family, is a widely distributed family of angiosperms that consists of 236 genera with more than 7000 species, which inhabit diverse ecosystems and have great diversity with a cosmopolitan distribution. The largest genera that belong to this family are *Salvia*, *Scutellaria*, *Stachys*, *Ajuga*, *Plectranthus*, *Hyptis*, *Teucrium*, *Vitex*, *Thymus*, and *Nepeta*. Many of them are used as spices and vegetables. The family has several species of aromatic plants that are applied in traditional medicine and in the pharmaceutical and food industries due to their biological properties. They act as a stimulant for blood circulation and digestion, as strengtheners of the central nervous system, and as expectorants, antispasmodics, antiseptics, diuretics, carminatives, and tonics [20]. *Lauraceae* is a large family of woody plants (except for the genus *Cassytha,* which is herbaceous) composed of 50 genera and 2500–3000 species. The distribution center of modern *Lauraceae* is in tropical to subtropical areas, including Southeast Asia, the Mediterranean, and Central and South America. The economic importance of this family lies in that many species are used in industrial sectors such as the food, timber, pharmaceutical, and perfumery industries. *Lauraceae* plants have been used as analgesics and applied to multiple different pathologies, such as the treatment of infectious diseases, malaria, gastrointestinal infections, female genital infections, and rheumatism [21]. *Myrtaceae* is one of the most important plant families, being regarded as the eighth-largest flowering plant family. It includes many genera of utmost ecological and economic importance distributed all over the world. Fruits are berry-type and have desirable pulp yield and organoleptic characteristics, as well as nutritional aspects and significant amounts of phytochemicals, such as organic acids, sugars, vitamins, polysaccharides, polyphenols, and important minerals. They can inactivate reactive oxygen species in the human body, thus promoting the risk reduction of various chronic diseases such as heart disease, stroke, cancer, atherosclerosis, diabetes, Alzheimer disease, cataracts, and pulmonary disorders [22]. *Rutaceae* comprises a large (about 160 genera and 1900 species), morphologically diverse, cosmopolitan family. The members are famous among phytochemists for their extraordinary array of secondary chemical compounds such as limonoids, flavonoids, coumarins, volatile oils, and alkaloids, many of which exhibit medicinal, antimicrobial, insecticidal, or herbicidal properties [23].

## 2. Materials and Methods

### 2.1. Bacterial Strains and Growth Conditions

This study was carried out on *C. violaceum* (ATCC 12742), *P. aeruginosa* (ATCC-9027), and *E. faecalis* OG1RF (ATCC 47077) strains. *C. violaceum* and *P. aeruginosa* were cultured in Nutrient broth containing 3 g/L of Beef extract and 5 g/L of Peptone at 30 °C. Meanwhile, *E. faecalis* was cultured in 36.4 g/L of Todd–Hewitt broth (THB) (Oxoid, Basingstoke, Hampshire, United Kingdom) at 37 °C under gentle agitation. The bacterial strains were kept frozen in stock cultures at −80 °C in cryovials.

### 2.2. Essential Oils (EOs) and Their Extraction

Ten EOs, provided from commercial producers located in Calabria (Southern Italy), were used. They were extracted from the following plants: *Clinopodium nepeta* L. (Kuntze, Carl Ernst Otto)*, Citrus bergamia* (Risso, Joseph Antoine & Poiteau, Pierre Antoine), *Citrus limon* L. (Osbeck, Pehr), *Citrus reticulata* (Blanco, Francisco Manuel)*, Foeniculum vulgare* subsp. *piperitum*, *Laurus nobilis* L., *Myrtus communis* L., *Origanum vulgare* L. subsp. *viridulum* (Martrin-Donos, Julien Victor) Nyman, Carl Frederik, *Salvia officinalis* L., and *Salvia rosmarinus* (Spenn, Fridolin Carl) [24].

The fruit peel of *C. bergamia* and *C. limon*, the flower, leaf, and terminal branches of *C. nepeta*, *F. vulgare*, *M. communis*, *O. vulgare*, *S. officinalis*, and *S. rosmarinus* were used, and only the leaf and terminal branches of *C. reticulata* and *L. nobilis* were used [12]. 

The EO from *C. bergamia* was mechanically extracted through an industrial cold expression process from fresh fruit. For all other species, EOs were extracted through the water-vapor-under-vacuum distillation process in a 20 L inox apparatus, starting with fresh collected raw material. The EOs were aliquoted and kept in dark glass bottles, tightly sealed at +4 °C, until use [12]. 

### 2.3. GC-MS Analysis of Essential Oils

Gas Chromatography-Mass Spectrometry (GC-MS) was performed using a gas chromatograph (Focus GC-Thermo Fisher, Milan, Italy) equipped with a Varian VF-5m (30 m × 0.25 mm × 0.25 μm) capillary column, combined with a single quadrupole mass spectrometer (DSQII-Thermo Scientific, Milan, Italy) [12]. The samples were diluted 1:1000 in ether. One microliter of the sample was injected in spitless mode at a temperature of 220 °C. The column flow rate was 1 mL min^-1^ using helium as carrier gas. The initial GC oven temperature was 55 °C, increased by 4 °C min^−1^ to 240 °C with a hold time of 3 min. The transfer line temperature was 250 °C. The MS was operated using electron impact (EI) at an ionization energy of 70 eV. The ion source temperature was set at 250 °C. The solvent delay for the mass spectrometry was set at 3 min and the EI scan mode was used for identification, covering the range of 25–350 *m/z*. The compound was identified by comparison with the NIST database (https://www.nist.gov/pml/atomic-spectra-database, accessed on 6 June 2021). The instrumentation performance, chromatograms, mass spectra, and initial data processing were carried out with the supplied Xcalibur software (Thermo Fisher, Milan, Italy).

### 2.4. Determination of Minimum Inhibitory Concentration (MIC) and Minimum Bactericidal Concentration (MBC)

Minimum inhibitory concentration (MIC) and minimum bactericidal concentration (MBC) for the ten EOs were determined using the broth dilution method. Since EOs do not dissolve in water or water-based mediums, in our study, to increase their dispersion in the aqueous medium, each EO was pre-absorbed on inulin, a fructan composed of a linear chain of connected fructose that is used in the food industry as a gelling agent and as a stabilizer for oil in water emulsions. Therefore, EOs were prepared daily by letting the oil adsorb to inulin powder to obtain a 100 µL EOs/gr inulin working solution, as previously described [12]. Approximately 10^9^ cells of *C. violaceum*, *P. aeruginosa*, and *E. faecalis* from an overnight culture of the two bacterial strains were inoculated into tubes containing 3 mL of the following serial dilutions of the dissolved EOs: 0.1, 0.2, 0.3, 0.4, 0.5, 0.6, 0.7, 0.8, 0.9, 1, 2, 3, 4, 5, 6, 7, 8, 9, 10 µL of EO/mL of medium. Culture tubes were shaken at 300 rpm at 37 °C for 18 h. In all experiments, a series of controls were set up: (1) *C. violaceum*, *P. aeruginosa*, or *E. faecalis* cells were inoculated into tubes containing liquid growth medium in absence of EOs to verify the optimal growth conditions (positive control); (2) liquid growth medium without inoculated cells and EOs to verify no growth of microorganisms (negative control), also providing evidence of sterility and lack of contamination; (3) *C. violaceum*, *P. aeruginosa*, or *E. faecalis* cells were inoculated into tubes containing liquid growth medium in the presence of the different inulin dilutions (vehicle control) to evaluate the potential effect of the substance on bacterial growth; (4) liquid growth medium without inoculated cells in presence of each EO to discern the turbidity background.

Turbidity measurement was performed at 600 nm in a spectrophotometer. The MIC values were determined as the lowest concentration of EO corresponding to optical density (OD) values comparable to those of the cell-free growth medium. The MBC values were calculated by subculturing all dilutions carried out in a liquid medium on agar plates. The MBC was determined by considering the lowest concentration of EO that reduces the viability of the initial bacterial inoculum by ≥99.9%. Each experiment was carried out in quintupled, with three independent repetitions.

### 2.5. Biofilm Formation Evaluation

Approximately 10^7^ cells of *C. violaceum* and *E. faecalis* from an overnight culture were inoculated into 96-well polystyrene microtiter plates containing 200 µL of growth medium in the absence and presence of the dissolved EOs at concentrations equal to and lower than those of MIC values. After 18 h of incubation, the planktonic cells were removed, and adherent cells were fixed with methanol and then stained with crystal violet solution 0.2%. After 5 min, the excess of the stain was removed by three repeated washes. Acetic acid at a concentration of 33% was added to each well and the optical density at 570 nm was determined spectrophotometrically. Each strain was examined in duplicate on each plate and the experiments were performed in quintupled. Positive and negative controls containing only the inoculated growth medium and the growth medium supplemented with EOs, respectively, were run in each experiment.

### 2.6. Violacein Production Evaluation

Approximately 10^9^ cells from an overnight culture of *C. violaceum* were inoculated into tubes containing 3 mL of Nutrient medium in the absence and presence of the dissolved EOs at concentrations equal to and lower than those of MIC values and incubated at 30 °C for 18 h. Then, 1 mL of the above culture was collected and centrifuged at 4 °C for 20 min. The supernatant was removed, and the violacein was dissolved in DMSO at room temperature. After a 3-min centrifugation to remove the cells, violacein production was spectrophotometrically quantified by reading the absorbance at 585 nm. Each sample was examined in duplicate in each plate and the experiments were performed in quintupled.

### 2.7. Gelatinase Activity Evaluation

Approximately 10^9^ cells from an overnight culture of *E. faecalis* were inoculated into tubes containing 3 mL of THB in the absence and presence of the dissolved EOs at concentrations equal to and lower than those of MIC values and incubated at 37 °C for 18 h. Then, 40 µL of culture supernatant was collected and incubated for 2 h with constant mixing (170 rpm) in the presence of 500 µL of azocoll substrate (10 mg/mL). After centrifugation at 20,000× *g* for 10 min, the protease activity of supernatant fractions was assayed spectrophotometrically by reading the absorbance at 550 nm. Each assay was performed in quintupled.

### 2.8. Genomic DNA Extraction

Genomic DNA was extracted from untreated *C. violaceum* and *E. faecalis* cells and cells treated with pre-inhibitory concentrations (pre-MICs) of EOs by using a DNeasy UltraClean Microbial Kit (Qiagen, Milan, Italy), according to the manufacturer’s protocol. Briefly, 3 mL of bacterial cell culture was collected and centrifuged for 10 min. The isolated pellets were resuspended in 300 µL of PowerBead Solution and transferred to PowerBead Tubes. An amount of 50 µL of Solution SL was added to each sample and, after vortexing for 10 min, the tubes were centrifuged at 10,000× *g* for 30 s. Supernatants were incubated at 4 °C for 5 min in the presence of 100 µL of Solution IRS and centrifugated at 10,000× *g* for 1 min. After the addition of 900 µL of Solution SB to the supernatants, 700 µL of the sample was loaded into MB Spin Columns and centrifuged at 10,000× *g* for 30 s. The centrifugation was repeated after adding 300 µL of Solution CB, and the flow-through was discarded. DNA samples were eluted by centrifugation at 10,000× *g* for 30 s in 50 µL of Solution EB. The DNA concentration and purity were determined spectrophotometrically by using the 260/280 nm absorbance ratio.

### 2.9. Quantification of Global 5-Methylcytosine Levels

Global DNA methylation levels of 5-methylcytosines (5mC) were determined by using the MethylFlash Global DNA Methylation (5mC) ELISA Easy Kit (Epigentek, Farmingdale, Nassau County, NY, USA), following the manufacturer’s instructions. Briefly, 100 ng of genomic DNA were mixed with 100 µL of Binding Solution, loaded in duplicate in a 96-well plate and then incubated at 37 °C for 60 min. After three consecutive washes of the wells with 150 μL of Wash Buffer, samples were incubated at room temperature for 50 min with 50 µL of 5-mC Detection Complex Solution, containing 5-mC antibody (1:1000), Signal Indicator (1:1000), and Enhancer Solution (1:2000). Then, the solution was removed from the wells through five consecutive washes with 150 μL of Wash Buffer, and 100 μL of Developer Solution was added to each well and incubated at room temperature for 3 min. The colorimetric reaction was stopped by the addition of 100 µL of Stop Solution and the absorbance was read at 450 nm using an ELISA plate reader. In each experiment, the percentage of 5-mC was calculated using the second-order regression equation of the standard curve obtained by diluting a positive control, provided by the kit, at different methylation percentages. The methylation values of *C. violaceum* and *E. faecalis* untreated cells were used as reference values (relative quantification) for the corresponding cells treated with the EOs.

### 2.10. Quantification of Global N6-Methyladenosine Levels

Global DNA methylation levels of N6-methyladenosines (m6A) were determined by using the MethylFlash m6A DNA Methylation ELISA Kit (Epigentek, Farmingdale, Nassau County, NY, USA), following the manufacturer’s instructions. Shortly, 200 ng of genomic DNA was mixed with 80 µL of Binding Solution, loaded in duplicate in a 96-well plate, and then incubated at 37 °C for 90 min. After three consecutive washes of the wells with 150 μL of Wash Buffer, samples were incubated at room temperature for 60 min with 50 µL of Capture Antibody. Then, the wells were washed thrice with 150 μL of Wash Buffer and incubated at room temperature for 3 min with 50 µL of Detection Antibody. After four consecutive washes, 50 μL of Enhancer Solution was added to each well, and the plate was incubated at room temperature for 30 min. Then, after five consecutive washes, wells were incubated at room temperature in the dark with 100 µL of Developer Solution. The colorimetric reaction was stopped by the addition of 100 µL of Stop Solution, and the absorbance was read at 450 nm using an ELISA plate reader. In each experiment, the percentage of m6A was calculated using the second-order regression equation of the standard curve obtained by diluting a positive control at different methylation percentages, as suggested by the kit. Each sample was examined in duplicate in each plate and the experiments were performed in triplicate. The methylation values of *C. violaceum* and *E. faecalis* untreated cells were used as reference values (relative quantification) for the corresponding cells treated with the EOs.

### 2.11. Statistical Analysis

Statistical analyses were performed using SPSS 28.0 statistical software (SPSS Inc., Chicago, IL, USA). Kruskal–Wallis one-way analysis of variance and Mann-Whitney tests were adopted. Significance level was defined as *p* ≤ 0.05.

## 3. Results

### 3.1. Characterization of the Essential Oils

Ten EOs were selected for the present study. Their characteristics in terms of family, chemical constituents, and their percentages and Retention Indices are presented in Table 1. Piperitone oxide, (+)-sabinene, estragole, and p-thymol represent the predominant compounds in *C. nepeta*, *C. reticulata*, *F. vulgare*, and *O. vulgare* oils, respectively. Eucalyptol is the major component of *L. nobilis*, *M. communis*, *S. officinalis*, and *S. Rosmarinus*. *C. bergamia* and *C. limon* Oils are characterized by a high content of (+)-limonene.

### 3.2. Effects of EOs on Cell Growth

The antibacterial activity of the EOs on the Gram-negative *C. violaceum and P. aeruginosa* and on the Gram-positive *E. faecalis* was evaluated by determining the MIC and the MBC values. Obtained results are reported in Table 2. Findings regarding *P. aeruginosa* are not reported as no variation in cell growth was observed at any EO concentration we used. For this reason, we then conducted the subsequent experiments only on *C. violaceum* and *E. faecalis.*

The analysis revealed that all EOs show bactericidal activity, as deduced by MIC values and confirmed by MBC. The EO from *O. vulgare* showed the greatest inhibitory effect on bacterial growth. The lowest MIC values, equal to 0.4 to 0.6 µL/mL in *C. violaceum* and *E. faecalis*, respectively, were identified in both bacterial strains. Under these MIC values, we also observed that the antibacterial activity of EOs from *F. vulgare* and *Clinopodium nepeta* appeared strain-specific since very low concentrations of these EOs are sufficient to inhibit the growth of *C. violaceum* (0.4 and 0.8 µL/mL, respectively). In comparison, higher concentrations of the two EOs are needed to inhibit that of *E. faecalis* (3 and 4 µL/mL, respectively). The vehicle did not show significant variations with respect to the untreated sample under any experimental conditions tested (data not shown).

Intermediate MIC values, corresponding to concentrations of 1 and 2 µL/mL, were observed following treatment with *L. nobilis* in *C. violaceum* and *E. faecalis*, respectively. The antibacterial activity of the remaining EOs is less effective, as evidenced by the higher MIC values, ranging from 2 to 5 µL/mL. Differences related to the bacterial strain are again observed, with *E. faecalis* appearing more resistant than *C. violaceum* to the inhibitory effects exerted by the EOs from *M. communis*, *S. officinalis,* and *S. rosmarinus*, and more sensitive to the EOs from *C. limon* and *C. reticulata.*

### 3.3. Effects of EOs on Biofilm Formation

The potential effect of the ten EOs on influencing biofilm formation was evaluated by the crystal violet method in *C. violaceum* and *E. faecalis,* kept in culture in the absence and presence of each EOs at concentrations equal to and lower than those of MIC values. The action of each EO against biofilm formation is shown in Figure 1 and Figure 2.

The results showed that all EOs had considerable anti-biofilm-forming effects on both *C. violaceum* (Figure 1) and *E. faecalis* (Figure 2). Indeed, by comparing the cells before and after treatment with the EOs, we can observe that they induced a general and progressive decreased ability to form biofilms (*p* < 0.001), with some exceptions. Specifically, in *E. faecalis*, treatment with concentrations of EOs from *C. limon* up to 0.6 induced an increase in biofilm formation (*p* < 0.001), with respect to the untreated strain. In contrast, higher concentrations lead to a progressive decline (*p* < 0.001). Moreover, a progressive increase in biofilm formation was observed following treatment, with *C. nepeta* concentrations ranging from 0.3 to 0.6 µL/mL (*p* = 0.016), followed by a dose-dependent decrease at higher concentrations of the EO (*p* < 0.001).

### 3.4. Effects of EOs on Quorum Sensing

The ability of EOs to interfere with the QS mechanism was determined by measuring the amount of violacein in *C. violaceum* and the gelatinase activity in *E. faecalis*, both influenced by the above process. We observed that the sub-inhibitory concentrations of the ten EOs show significant concentration-dependent inhibition of violacein production in *C. violaceum* (*p* < 0.001), which depends on the QS regulation (Figure 3). Consistently, the action of sub-inhibitory concentrations of all EOs appears effective in generally reducing the gelatinase activity in *E. faecalis* (Figure 4), with slight differences among the different EOs.

A significant and progressive decrease in the enzymatic activity of gelatinase was observed as a function of the increasing concentration of *C. nepeta*, *C. bergamia*, and *S. officinalis* essential oils (*p* < 0.001). An approximately 80% decline in gelatinase activity was observed following the treatment with 0.1 µL/mL of *F. vulgare*, *L. nobilis*, *M. communis* and *O. vulgare*. Then the activity remained constant or further decreased under high concentrations. This activity showed a significant increase following treatment with concentrations of *C. reticulata* between 0.3 and 1 µL/mL (*p* = 0.032), and then a drastic decrease at higher concentrations (*p* = 0.008). Concentrations of *C. limon* between 0.1 and 0.4 µL/mL did not significantly influence a drastic drop in the activity of gelatinase (*p* = 0.008) that underwent an increase of 30% following concentrations between 0.6 and 2 µL/mL (*p* = 0.008). Lastly, concentrations of *S. rosmarinus* between 0.1 and 0.2 µL/mL significantly increased gelatinase activity by 40% (*p* = 0.008), which was later reduced to higher concentrations (*p* < 0.001).

### 3.5. Effects of EOs on DNA Methylation Profiles

Global methylation levels of cytosine and adenine residues were evaluated in DNA samples extracted from *C. violaceum* and *E. faecalis* kept in culture in the absence and presence of each EO. Figure 4 shows the abundance of 5-methylcytosine (5mC) and N6-methyladenosine (m6A) in *C. violaceum*. By comparing the cells before and after treatment with the EOs at sub-inhibitory concentrations, we can observe that they induced a significant reduction in the methylation status of cytosine residues (*p* < 0.001) (Figure 5A). Similarly, a general down methylation of the adenine residues was detected (*p* < 0.05), with some exception (Figure 5B). No significant change was observed in the methylation status of adenine under the sub-inhibitory concentration of *C. limon*, *F. vulgare* or *O. vulgare*.

In *E. faecalis*, EOs treatment resulted in much more variable levels of 5mC and m6A (Figure 6). An increase in 5mC levels was observed with *C. limon*, *C. reticulata*, *F. vulgare*, *L. nobilis*, *S. officinalis*, and *S. rosmarinus* (*p* < 0.01). Meanwhile, a decrease was noticed for the sole *O. vulgare* (*p* < 0.001). No significant change was observed after treatment with *C. nepeta*, *C, bergamia*, or *M. communis* (Figure 6A).

Lastly, a general reduction in m6A was detected (*p* < 0.01), with some exceptions (Figure 5B). No significant change was observed with *C. bergamia*, *S. officinalis*, and *S. rosmarinus* (Figure 6B).

## 4. Discussion

The largely unregulated use of antibiotics has increased the prevalence of antibiotic-resistant bacteria. Therefore, interest in natural medicine has been growing, prompting the search for valid alternatives to non-organic synthetic antibiotic compounds. In this context, natural products, such as essential oils extracted from plants, are regarded as an important source of new molecules for drug and therapy discovery [25,26]. The historical use of EOs as antimicrobial agents has long been well-known, but only recently have a variety of experimental data demonstrated their bioactive properties in-depth. These properties can be explained through the induction of irreversible damage to the bacterial cell wall and membrane, the inhibition of metabolic pathways and protein synthesis, and interference with cell wall synthesis and DNA and RNA synthesis, according to many groups of chemical compounds which comprise them [8,27,28].

In a previous study, we reported that the essential oils extracted from *C. nepeta*, *C. bergamia*, *C. limon*, *C. reticulata*, *F. vulgare, L. nobilis*, *M. communis*, *O. vulgare*, *S. officinalis*, and *S. rosmarinus* aromatic plants show antibacterial activity against an *E. coli* strain and three lines derived by growing it at low concentrations of ampicillin, ciprofloxacin, and gentamicin [12]. Here we want to investigate whether the antibacterial properties of these oils are also effective against other strains, thus establishing that the inhibition of microbial growth mediated by the EOs is a generalized and non-strain-specific phenomenon. To this purpose, the study was conducted on Gram-negative *C. violaceum* and *P. aeruginosa*, and Gram-positive *E. faecalis*, commonly used for years to identify and characterize bioactive substances acting against microorganisms. *C. violaceum* has recently emerged as an important model of an environmental pathogen responsible for opportunistic infections that quickly lead to the host’s death. Its high pathogenicity in humans and animal models of infection involves the possession of several predicted virulence traits, including secreted chitinase and chitosanase proteins as well as proteins involved in the transport and capture of amino acids and carbohydrates, in addition to oxidative stress protection [29]. *P. aeruginosa* is found in soil, water, and animals, and it is an opportunistic pathogen in humans. It infects the pulmonary and urinary tracts, wounds, and burns, thus causing both severe acute and chronic infections. It is particularly critical in healthcare-associated infections and in ill and immunocompromised patients [30]. Enterococci are resilient and versatile bacteria capable of surviving in hostile environments. They have gained particular significance for human health, being the principal cause of hospital-acquired or healthcare-associated infections affecting the urinary tract, bloodstream, wounds, and the endocardium [31]. Particularly, *E. faecalis*, characterized by antibiotic resistance traits and biofilm formation capabilities, was also found to occur as a food contaminant and in secondary endodontic biofilm-associated infections [32].

Although *E. feacalis* appeared more resistant to oil stimulation, showing MIC values mostly higher than those observed in *C. violaceum*, the results demonstrated that all essential oils possess antibacterial activity. The EO extracted from *O. vulgare* was the most effective in inhibiting the growth of the two bacterial strains, with a minimum MIC of 0.4 and 0.6 µL/mL. Additionally, EOs from *C. nepeta* and *F. vulgare* had strong antibacterial activity against *C. violaceum*. This evidence agrees with previous studies, which reported that these EOs have strong antimicrobial activity against other bacterial and fungal species, including *E. coli* JM109 and its derived antibiotic-resistant cells, as well as *S. aureus*, *Listeria monocytogenes*, *Salmonella typhimurium*, *Penicilium chrysogenum*, *Alternaria alternata*, *and Chaetomium globosum* [12,33,34,35,36,37,38]. The antimicrobial effect of some monoterpenes against Gram-positive and Gram-negative bacterial strains seems to be achieved through a perturbation of the lipid fraction of bacterial plasma membranes, resulting in alterations of membrane permeability and leakage of intracellular materials [39]. No growth inhibition at the concentrations of essential oils we used was observed in *P. aeruginosa*. This result is substantially in line with literature data, which reported that this organism is less susceptible than other species to treatment with different essential oils, or is susceptible only to high concentrations [40,41]. In our work, we had to limit the analysis to concentrations equal or less than 10 µL/mL since the EOs do not completely dissolve in inulin at higher concentrations. In any case, once again *P. aeruginosa* demonstrates greater resistance to molecules with antimicrobial activity than other bacterial species.

Since biofilm formation is one of the major protective mechanisms for survival of microbes under adverse conditions, we investigated whether the antibacterial effect of the EOs was associated with a reduction in the growth ability of *C. violaceum* and *E. faecalis* in biofilm structures. Sub-inhibitory concentrations of the EOs, which were insufficient to inhibit the bacterial growth, were used for evaluating their potential antibiofilm activity. Results revealed that all EOs significantly induced a dose-dependent inhibition of the biofilm formation in both bacterial strains. Thymol and terpinene are the major components characterizing *O. vulgare* oil. Previous data reported that the antimicrobial effect of some monoterpenes against Gram-positive and Gram-negative bacterial strains may be due to a perturbation of the lipid fraction of bacterial plasma membranes, resulting in alterations to membrane permeability and in leakage of intracellular materials [42]. Therefore, the anti-biofilm-forming property of this oil is consistent with results showing that these two compounds may alter the permeability of plasma membranes, disrupt bacterial adhesion to the surfaces, and subsequently reduce biofilm formation by *Staphylococcus epidermidis*, *Bacillus cereus*, *Pichia anomala,* and *Pseudomonas putida* [12,19,42]. Similarly, the strong antimicrobial and antibiofilm efficacy of *F. vulgare* oil is consistent with the results recently observed in other bacterial strains and fungi, and suggests that it is strictly influenced by the massive presence of phenylpropanoids (estragole and anethale) and monoterpenes (α-pinene and α-phellandrene) [43,44]. It has been reported that EOs characterized by a high level of piperitenone and piperitone, such as, for example, the one obtained from *C. nepeta*, significantly inhibit bacterial biofilm formation and the metabolism of both Gram-positive and Gram-negative bacterial strains [45].

The decreased ability to grow in biofilm could suggest a quencher role exercised by the EOs on the QS process by inhibiting the activity of receptors and molecules involved in this pathway. Finding a significant reduction of violacein production and gelatinase activity, two widely recognized biomarkers of the QS process in *C. violaceum* and *E. faecalis*, respectively, suggests that the ten EOs analyzed interfere with the QS system and inhibit cellular communication pathways, thus preventing the assembly of the biofilm matrix.

We previously demonstrated that epigenetic modifications mediate the antibacterial properties of the EOs since they induce a profound remodeling of the methylation levels of adenine and cytosine residues in the genomes of *E. coli* and its derived antibiotic-resistant cells [12]. Results we observed in *C. violaceum* and *E. faecalis* corroborated this evidence, demonstrating that it is a generalized phenomenon not linked to a specific bacterial strain. In most cases, we observed down methylation at both cytosine and adenine residues after treatment with EOs, with more marked changes occurring in the *C. violaceum* genome. Whether these changes are correlated with the inhibition or activation of gene expression deserves further investigation.

It is plausible that a massive epigenetic remodeling at the DNA level is responsible for an equal modulation of the expression of genes involved in various processes—including cooperative behaviors, cell-to-cell communication, and virulence—that are beneficial when performed by groups of bacteria acting in synchrony. Through these mechanisms, the EOs would probably hinder the strategies adopted for the survival of the bacterial community, thus inhibiting bacterial growth.

Overall, this study reports, for the first time, that different EOs induce epigenetic changes and negatively regulate biofilm formation and the QS mechanism, providing further characterization at the molecular level of the antimicrobial properties of EOs. The antibacterial, anti-biofilm, and anti-QS efficacy of the tested EOs, especially those extracted from *O. vulgare* and *F. vulgare*, appear to be of relevance since these oils may represent effective and natural therapeutic compounds that can find applications, either alone or in combination with traditional antibiotics, in the control of bacterial infections in countless fields.

## Figures and Tables

**Figure 1 microorganisms-11-01150-f001:**
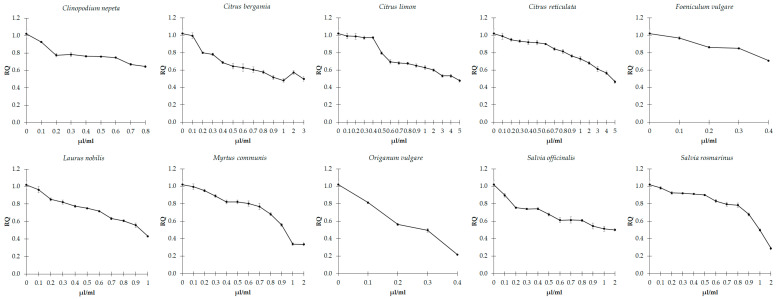
Effect of sub-inhibitory concentrations of essential oils on biofilm formation in *C. violaceum*. Values are reported as relative quantification (RQ), determined using the untreated cells as reference. Values represent the mean of five independent duplicate experiments with standard error of the mean.

**Figure 2 microorganisms-11-01150-f002:**
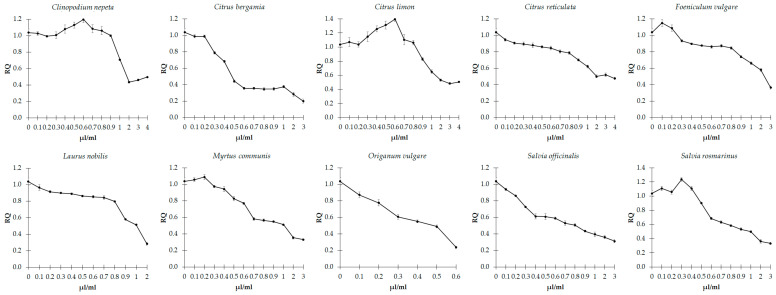
Effect of sub-inhibitory concentrations of essential oils on biofilm formation in *E. faecalis*. Values are reported as relative quantification (RQ), determined using the untreated cells as reference. Values represent the mean of five independent duplicate experiments with standard error of the mean.

**Figure 3 microorganisms-11-01150-f003:**
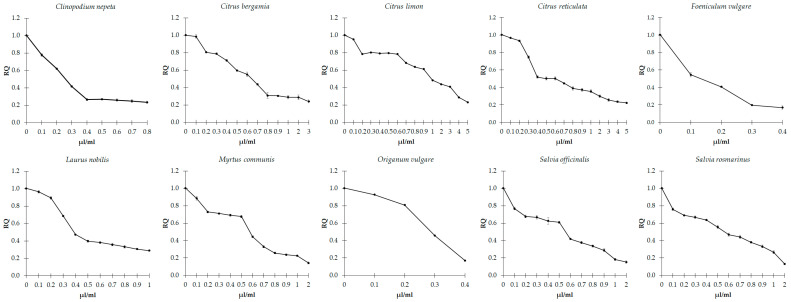
Effect of sub-inhibitory concentrations of essential oils on violacein production in *C. violaceum*. Values are reported as relative quantification (RQ), determined using the untreated cells as reference. Values represent the mean of five independent duplicate experiments with standard error of the mean.

**Figure 4 microorganisms-11-01150-f004:**
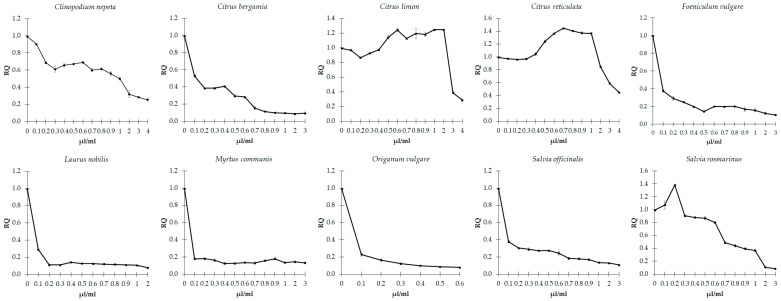
Effect of sub-inhibitory concentrations of essential oils on the gelatinase activity in *E. faecalis*. The values are reported as relative quantification (RQ), determined using the untreated cells as reference. Values represent the mean of five independent duplicate experiments with standard error of the mean.

**Figure 5 microorganisms-11-01150-f005:**
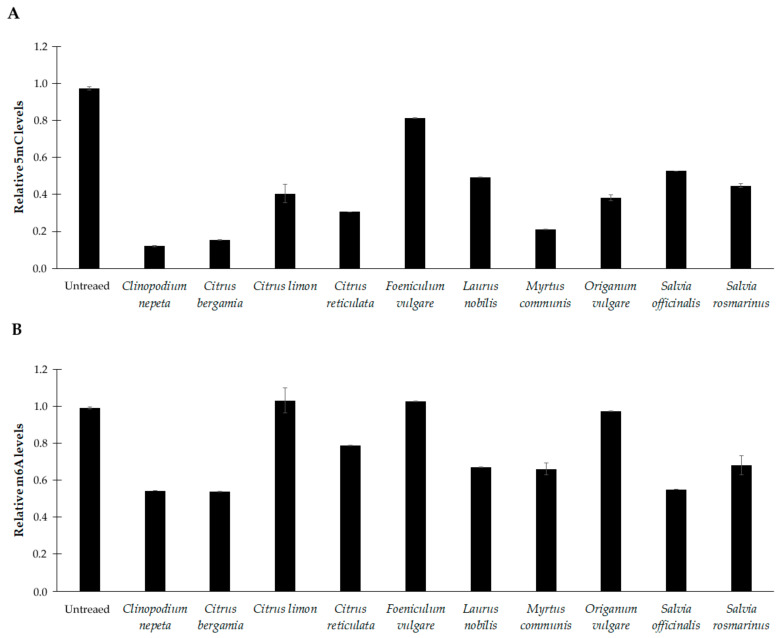
Methylation levels of 5-methylcytosine (5mC) (**A**) and N6-methyladenosine (m6A) (**B**) residues in DNA samples extracted from *C. violaceum* at basal conditions (untreated) and after treatment with sub-inhibitory concentrations of the essential oils. The values are reported as relative quantification (RQ), determined using the values of the untreated cells as reference. Values represent the mean of five independent triplicate experiments with standard error of the mean.

**Figure 6 microorganisms-11-01150-f006:**
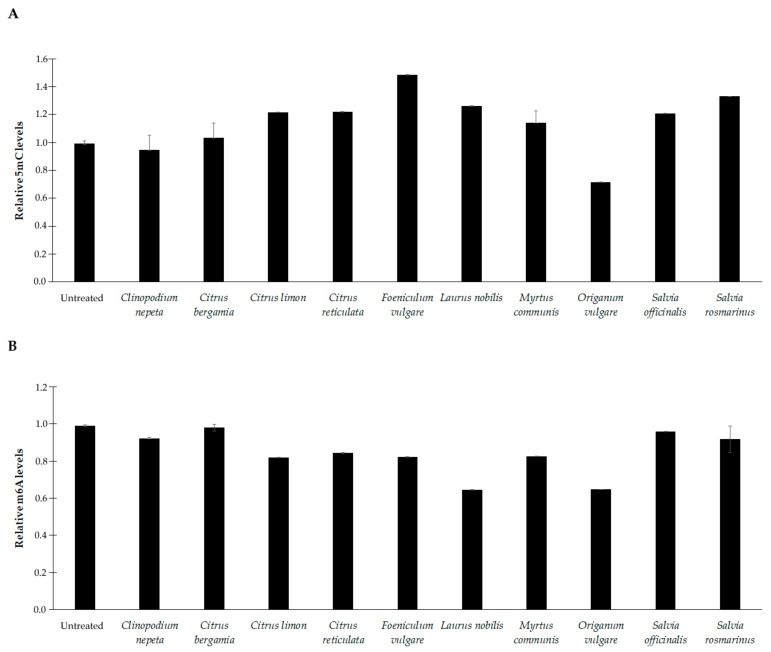
Methylation levels of 5-methylcytosine (5mC) (**A**) and N6-methyladenosine (m6A) (**B**) residues in DNA samples extracted from *E. faecalis* at basal conditions (untreated) and after treatment with sub-inhibitory concentrations of the essential oils. The values are reported as relative quantification (RQ), determined using the values of the untreated cells as reference. Values represent the mean of five independent triplicate experiments with standard error of the mean.

**Table 1 microorganisms-11-01150-t001:** Characterization of the essential oils analyzed (presence ≥ 1%).

Essential Oil	Family	Extraction Method	Chemical Composition	%	Retention Index	Cultivation Area	Time of Collection	Batch N.
*Clinopodium nepeta*	*Lamiaceae*	Hydro distillation from fresh collected material	piperitone oxide	34.28	16.14	Briatico, Province of Vibo Valentia, Calabria Region, Southern Italy Michele Crudo Farm	July 2021	20210715
			piperitenone oxide	18.23	19.72			
			(+)-limonene	15.8	8.49			
			(+)-pulegone	13.75	15.6			
			menthone	8.32	12.77			
			isolegylacetate	3.64	17.92			
			1-terpine-4-ol	1.4	13.56			
			(+)-neomenthol	1.37	13.25			
			β-pinene	1.22	6.92			
*Citrus bergamia*	*Rutaceae*	Mechanical extraction through industrial cold expression process from fresh fruit	(+)-limonene	15.89	34.28	Bovalino, Province of Reggio Calabria, Calabria Region, Southern Italy Luigi Frammartino Farm	December 2021	20211221
			lynalyl acetate	10.78	11.54			
			(+)-linalool	9.38	6.79			
			α-terpinene	8.44	38.88			
			β-pinene	6.87	5.49			
			α-pinene	5.64	1.22			
*Citrus limon* (L.)	*Rutaceae*	Hydro distillation from fresh collected material	(+)-limonene	14.13	3.01	Bovalino, Province of Reggio Calabria, Calabria Region, Southern Italy Luigi Frammartino Farm	November 2021	2021118
			α-terpinene	13.59	1.26			
			β-pinene	10.28	1.67			
			α-terpineol	9.36	11.91			
			α-terpinolene	8.42	74.41			
			1-Terpine-4-ol	6.84	4.34			
*Citrus reticulata*	*Rutaceae*	Hydro distillation from fresh collected material	(+)-sabinene	12.6	1.44	Briatico, Province of Vibo Valentia, Calabria Region, Southern Italy Michele Crudo Farm	February 2021	20210215
			(+)-linalool	10.76	18.27			
			α-phellandrene	10.27	1.37			
			β-cis-ocimene	9.36	1.14			
			(+)-limonene	8.96	6.45			
			β-myrcene	8.42	5.04			
			β-pinene	7.76	6.54			
			α-pinene	7.16	2.37			
			β-citronellal	6.86	2.35			
			α-terpinolene	6.72	50.91			
			α-terpinene	5.63	1.93			
*Foeniculum vulgare* subsp. *piperitum*	*Apiaceae*	Hydro distillation from fresh collected material	estragole	17.21	14.54	Briatico, Province of Vibo Valentia, Calabria Region, Southern Italy Michele Crudo Farm	September 2021	20210918
			α-pinene	14.19	45.33			
			anethal	10.41	11.24			
			fenchone	8.42	8.49			
			α-limonene	7.72	2.51			
			α-phellandrene	7.18	1.05			
			β-pinene	6.86	1.65			
			β-myrcene	5.63	14.71			
*Laurus nobilis* L.	*Lauraceae*	Hydro distillation from fresh collected material	eucalyptol	21.07	1.51	Briatico, Province of Vibo Valentia, Calabria Region, Southern Italy Michele Crudo Farm	April 2021	20210413
			(+)-sabinene	19.18	6.48			
			(+)-linalool	13.6	1.29			
			terpinyl acetate	10.79	7.38			
			α-pinene	8.56	56.61			
			methyleugenol	6.73	15.74			
			1-terpine-4-ol	5.64	5.65			
*Myrtus communis* L.	*Myrtaceae*	Hydro distillation from fresh collected material	eucalyptol	20.27	1.88	Briatico, Province of Vibo Valentia, Calabria Region, Southern Italy Michele Crudo Farm	May 2021	20210521
			(−)-myrtenylacetate	18.43	17.04			
			α-pinene	15.92	3.88			
			(+)-limonene	14.16	2.1			
			(+)-linalool	10.82	10.43			
			lynalyl acetate	10.33	1.1			
			geraniol acetate	9.44	1.09			
			α-terpineol	9.04	1.58			
			β-ocimene	8.59	33.04			
			α-phellandrene	8.49	10.81			
			o-cymene	8.35	1.41			
			terpinolene	7.79	1.41			
			terpinene	5.69	12.33			
*Origanum vulgare* L. subsp. *viridulum*	*Lamiaceae*	Hydro distillation from fresh collected material	p-thymol	20.48	4.88	Briatico, Province of Vibo Valentia, Calabria Region, Southern Italy Michele Crudo Farm	July 2021	20210720
			terpinene	16.77	47.31			
			p-cymene	16.42	3.52			
			β-caryophyllene	8.63	18.52			
			β-myrcene	7.61	11.78			
			carvacrol	7.36	3.18			
			terpinolene	6.55	3.76			
			α-thujene (origanene)	5.15	1.23			
			α-pinene	4.96	2.73			
*Salvia officinalis* L.	*Lamiaceae*	Hydro distillation from fresh collected material	eucalyptol	22.6	5.54	Briatico, Province of Vibo Valentia, Calabria Region, Southern Italy Michele Crudo Farm	May 2021	20210504
			(−)-α-thujone	21.5	2.85			
			β-pinene	12.45	9.59			
			(−)-camphor	11.44	4.35			
			α-humulene	11.07	24.14			
			(−)-β-thujone	8.58	23.7			
			α-pinene	7.23	2.26			
			(−)-β-caryophyllene	6.9	15.1			
			β-myrcene	6.75	1.13			
			C camphene	6.12	1.88			
			(+)-sabinene	5.67	3.99			
*Salvia rosmarinus*	*Lamiaceae*	Hydro distillation from fresh collected material	eucalyptol	21.52	1.17	Briatico, Province of Vibo Valentia, Calabria Region, Southern Italy Michele Crudo Farm	April 2021	20210421
			α-pinene	13.31	2.28			
			β-pinene	12.45	3.66			
			camphene	8.56	49.29			
			(−)-camphor	7.21	1.79			
			isoborneol	6.89	9.26			
			β -myrcene	6.1	6.7			
			(−)-β-caryophyllene	5.66	22.84			

**Table 2 microorganisms-11-01150-t002:** Minimum inhibitory concentrations (MIC) and minimum bactericidal concentrations (MBC) expressed as µL/mL of essential oils against *C. violaceum* and *E. faecalis*. The values represent the mean of five independent triplicate experiments. SEM: standard error of the mean.

Essential Oils	*C. violaceum*				*E. faecalis*			
	MIC		MBC		MIC		MBC	
	Mean	SEM	Mean	SEM	Mean	SEM	Mean	SEM
*C. nepeta*	0.80	0.07	0.80	0.07	4.00	0.00	4.00	0.00
*C. bergamia*	3.00	0.00	3.00	0.00	3.00	0.00	3.00	0.00
*C. limon*	5.00	0.07	5.00	0.00	4.00	0.00	4.00	0.00
*C. reticulata*	5.00	0.07	5.00	0.00	4.00	0.00	4.00	0.00
*F. vulgare*	0.40	0.00	0.40	0.00	3.00	0.07	3.00	0.07
*L. nobilis*	1.00	0.00	1.00	0.00	2.00	0.07	2.00	0.07
*M. communis*	2.00	0.00	2.00	0.00	3.00	0.07	3.00	0.07
*O. vulgare*	0.40	0.00	0.40	0.00	0.60	0.00	0.60	0.00
*S. officinalis*	2.00	0.00	2.00	0.00	3.00	0.00	3.00	0.00
*S. rosmarinus*	2.00	0.00	2.00	0.00	3.00	0.00	3.00	0.00

## Data Availability

Research data is available upon request by contacting the corresponding author of the article.

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
