# Peer review of "Effect of Essential Oils of Apiaceae, Lamiaceae, Lauraceae, Myrtaceae, and Rutaceae Family Plants on Growth, Biofilm Formation, and Quorum Sensing in Chromobacterium violaceum, Pseudomonas aeruginosa, and Enterococcus faecalis"

_microorganisms, 2023, doi:10.3390/microorganisms11051150_

Round 1

Reviewer 1 Report

1 - In general, the article: Effect of Essential Oils on bacterial growth, biofilm formation, 2 and Quorum Sensing in Chromobacterium violaceum, Pseudomonas aeruginosa, and Enterococcus faecalis needs several  adjustments, described below:

2 – In title: Please put the family of the species in question in the title.

3 – Introduction: Put who described each species when citing the scientific name of the specie for the first time. Remember the rules of scientific naming Ex.: Origanum vulgare L.; Bacillus subtilis Cohn. Do this with all the species mentioned in first time.

4 – Introduction:  Talk a little about each species of plant worked on. This is essential to situate the reader in this article. Try to group the species by families and thus talk a little about each family.

5 - Materials, Lines 67-72, please enter the registration number of the identification specimen deposit. Please also place the record of inclusion of the genetic heritage in the database of the country where it was collected. Include the following plant collection information: Geographical coordinates, time of collection, month and year of collection.

6 - Materials, Lines 82-83: What would the positive control be? What would be the negative control. THAT INFORMATION WAS NOT VERY CLEAR. The positive control would be a drug or medicine already used commercially? The negative control would be the medium without any treatment?

7 - Materials, Lines 73-87: It is necessary to place a vehicle control. The vehicle control is the medium plus the diluent used (DMSO, Polyethylene glycol, acetone) to dissolve the essential oil in the medium. Remember that the essential oil does not dissolve in water or a liquid medium and that a diluent or surfactant is required.

8 - Materials, Lines 60-65 Bacterial strains were cultured in Nutrient broth. Authors have given no evidence of testing cultures for contamination (bacterial or fungi etc), please include all stringency to avoid such contamination

9 - Materials, Lines 18-183: How did you come to the conclusion of doing an ANOVA test or and Student’s t-test without first having done the test to verify whether the data have a normal distribution or not. What test was performed to verify the normality of the data? How many biological replicates were performed? Please include all this information in the topic: Statistics

10 - Materials, Analysis of Essential Oils and Fractions by GC-MS (IT IS NECESSARY TO PUT THIS TOPIC): It is necessary to inform how the analysis by gaseous coloratography was carried out. State which equipment was used for the analysis, which column was used, which carrier gas was used, the temperature conditions for sample infection, the temperature conditions of the program.

11 – Results, Table 1: Compound identity can be confirmed with an authentic standard or can be identified by comparing the RI and mass spectra to the Adams and Wiley databases. Table 1 does not bring this information. In addition, IT IS NECESSARY TO PUT THE OBTAINED RETENTION INDEX

12 – Results, Figure 1 and 2: Figure 1 and 2 are out of focus and the graphics are too small, the "x" axis values are practically difficult to see. Please fix this.

Author Response

1 - In general, the article: Effect of Essential Oils on bacterial growth, biofilm formation, 2 and

Quorum Sensing in Chromobacterium violaceum, Pseudomonas aeruginosa, and

Enterococcus faecalis needs several adjustments, described below:

2 – In title: Please put the family of the species in question in the title.

Response: following the reviewer’s concern, we put the family of the species in the title.

3 – Introduction: Put who described each species when citing the scientific name of the specie for the first time. Remember the rules of scientific naming Ex.: Origanum vulgare L.; Bacillus

subtilis Cohn. Do this with all the species mentioned in first time.

Response: following the reviewer’s concern, in the revised version of the manuscript, we indicated who described each specie and the exact scientific name as reported in the database International Plant Names Index (IPNI) (http://www.ipni.org).

4 – Introduction: Talk a little about each species of plant worked on. This is essential to situate the reader in this article. Try to group the species by families and thus talk a little about each family.

Response: in the revised version of the manuscript, we reported in the “Introduction” paragraph some information about the species of plants we analyzed.

5 - Materials, Lines 67-72, please enter the registration number of the identification specimen deposit. Please also place the record of inclusion of the genetic heritage in the database of the country where it was collected. Include the following plant collection information: Geographical coordinates, time of collection, month and year of collection.

Response: As stated in Acknowledgments and Conflicts of Interest declarations, the Essential Oils were sourced from local producers. Oils are part of the producer's commercial batches and do not belong to any botanical collections. From the lot number of each sample, the company has provided information relative to the reviewer’s concern about the place of collection and time of collection. In the revised version of the manuscript, we reported them in Table 1.

6 - Materials, Lines 82-83: What would the positive control be? What would be the negative control. THAT INFORMATION WAS NOT VERY CLEAR. The positive control would be a drug or medicine already used commercially? The negative control would be the medium without any treatment?

Response: Considering the reviewer's concerns, in the “Materials and Methods” paragraph of the revised version of the manuscript we better clarify the control samples included in each experimental condition.

7 - Materials, Lines 73-87: It is necessary to place a vehicle control. The vehicle control is the medium plus the diluent used (DMSO, Polyethylene glycol, acetone) to dissolve the essential oil in the medium. Remember that the essential oil does not dissolve in water or a liquid medium and that a diluent or surfactant is required.

Response: We are fully aware that essential oils do not dissolve per se in water or water-based medium. In fact, in our study, EOs were pre-absorbed on Inulin, a fructan composed of a linear chain of connected fructose that is used in the food industry as gelling agent and as a stabilizer for oil in water emulsion. In the “Materials and Methods” paragraph of the revised version of the manuscript, we detailed that serial dilution of inulin (vehicle) with inoculated cells were analysed, to evaluate the potential effect of the substance on bacterial growth. The vehicle did not show significant variations with respect to the untreated sample under any experimental conditions tested.

8 - Materials, Lines 60-65 Bacterial strains were cultured in Nutrient broth. Authors have given no evidence of testing cultures for contamination (bacterial or fungi etc), please include all stringency to avoid such contamination.

Response: We better described in the “Material and Methods” paragraph of the revised version of the manuscript, all controls carried out in each experimental phase. In particular, control 2 represented by a liquid growth medium without inoculated cells and EOs is utilized to verify no growth of microorganisms (negative control) and give also evidence of sterility and lacking of contamination.

9 - Materials, Lines 18-183: How did you come to the conclusion of doing an ANOVA test or and Student’s t-test without first having done the test to verify whether the data have a normal distribution or not. What test was performed to verify the normality of the data? How many biological replicates were performed? Please include all this information in the topic: Statistics

Response: The observation of the reviewer is correct. We performed the tests, as is reported in most of the studies of this kind, without reporting data on normality. On the other hand, as the reviewer is certainly suggesting, the normality test does not make much sense with a limited number of experiments (5). For this reason, in the revised version of the manuscript, we reported the number of replicas and the results of non-parametric tests.

10 - Materials, Analysis of Essential Oils and Fractions by GC-MS (IT IS NECESSARY TO PUT THIS TOPIC): It is necessary to inform how the analysis by gaseous coloratography was carried out. State which equipment was used for the analysis, which column was used, which carrier gas was used, the temperature conditions for sample infection, the temperature conditions of the program.

Response: the reviewer is right about the lack of the above information. We have underestimated their importance by putting in the manuscript only reference 12 relative to a paper we published in 2022 which contains them in detail. In the revised version of the manuscript, we introduced the information in the “Materials and Methods” paragraph.

11 – Results, Table 1: Compound identity can be confirmed with an authentic standard or can be identified by comparing the RI and mass spectra to the Adams and Wiley databases. Table 1 does not bring this information. In addition, IT IS NECESSARY TO PUT THE OBTAINED RETENTION INDEX

Response: following the reviewer’s concern, we have introduced the Retention Index in Table 1.

12 – Results, Figure 1 and 2: Figure 1 and 2 are out of focus and the graphics are too small, the "x" axis values are practically difficult to see. Please fix this

Response: we redid the figures considering the revisions introduced in the manuscript in response to the reviewer's comments and to increase their resolution.

Reviewer 2 Report

The paper is a good scientific work, dealing with the bactericidal activities of some EOs. Search for alternatives to synthetic antibiotics, e,g. those of plant origin,  is a new approach. The paper is well designed. There are some criticisms as follows:

Minor criticisms

- Scientific names should be italicized.

-Scientific names should be spelled out for the first time, then the genus name should be abbreviated.

-Abbreviations in abstract unless they were firstly defined should be avoided as abstract stands alone.

- In Figures 4 & 5, horizontal axis, do not italicize "Untreated".

- It is much better to refer to the families of the tested EOs.

The authors can be guided by the attached PDF file.

Major criticisms

- Three replicates are little for statistical analyses.

- Phytochemical profile of the tested EOs  is an essential part of the current study. Unfortunately, aim of the study and Materials & Methods lack this point.

-Characterization of EOs need to be clarified in detail in Results instead of being just Table 1. By the way, insert the Retention Time (RT) in this Table.

-It is well known that the major constituents (about 90%) of EOs are monoterpenes. However, the relation between the chemical constituents (or the main metabolite) of the tested EOs and the present results is missing.

Author Response

The paper is a good scientific work, dealing with the bactericidal activities of some EOs. Search for alternatives to synthetic antibiotics, e,g. those of plant origin,  is a new approach. The paper is well designed. There are some criticisms as follows:

Minor criticisms

- Scientific names should be italicized.

-Scientific names should be spelled out for the first time, then the genus name should be abbreviated.

-Abbreviations in abstract unless they were firstly defined should be avoided as abstract stands alone.

- In Figures 4 & 5, horizontal axis, do not italicize "Untreated".

- It is much better to refer to the families of the tested EOs.

The authors can be guided by the attached PDF file.

Response: in the revised version of the manuscript, we addressed all criticisms of the reviewer. In particular, we introduced the family plants in Table 1.

Major criticisms

- Three replicates are little for statistical analyses.

Response: following the comment of the reviewer, we added data from two more experiments.

- Phytochemical profile of the tested EOs  is an essential part of the current study. Unfortunately, aim of the study and Materials & Methods lack this point.

Response: the reviewer is right about the lack of the above information. We have underestimated their importance by putting in the manuscript only reference 12 relative to a paper we published in 2022 which contains them in detail. In the revised version of the manuscript, we introduced the information in the “Materials and Methods” paragraph.

-Characterization of EOs need to be clarified in detail in Results instead of being just Table 1. By the way, insert the Retention Time (RT) in this Table.

Response: We introduced, in the revised version of manuscript, a paragraph reporting the characterization of EOS. What is more, we added the Retention Index in Table 1.

-It is well known that the major constituents (about 90%) of EOs are monoterpenes. However, the relation between the chemical constituents (or the main metabolite) of the tested EOs and the present results is missing.

Response: following the comment of the reviewer, in the Discussion paragraph of the revised version of manuscript, we have introduced some comments on the relation between the chemical constituents of the Essential Oils and the results of our study relatively to the EOs which have exhibited the major antibacterial activity.

Reviewer 3 Report

Dear authors and editor.

I believe that the manuscript can be published after general editorial changes

Author Response

Dear authors and editor.

I believe that the manuscript can be published after general editorial changes.

Response: thanks to the reviewer for the comment. We have completely revised the manuscript taking into consideration the comments of the reviewers

Round 2

Reviewer 1 Report

The authors answered most questions and suggestions. After an extensive review by the journal's editorial board, the present version of the article can move on to the next steps for publication.